# The Overall Survival and Progression-Free Survival in Patients with Advanced Adrenocortical Cancer Is Increased after the Multidisciplinary Team Evaluation

**DOI:** 10.3390/cancers14163904

**Published:** 2022-08-12

**Authors:** Irene Tizianel, Mario Caccese, Francesca Torresan, Giuseppe Lombardi, Laura Evangelista, Filippo Crimì, Matteo Sepulcri, Maurizio Iacobone, Marta Padovan, Francesca Galuppini, Vittorina Zagonel, Carla Scaroni, Filippo Ceccato

**Affiliations:** 1Department of Medicine DIMED, University of Padova, 35128 Padua, Italy; 2Endocrine Disease Unit, University-Hospital of Padova, 35128 Padua, Italy; 3Department of Oncology, Oncology 1, Veneto Institute of Oncology IOV-IRCCS, 35128 Padua, Italy; 4Endocrine Surgery Unit, Department of Surgical, Oncological and Gastroenterological Sciences (DiSCOG), University of Padova, 35128 Padua, Italy; 5Nuclear Medicine Unit, University-Hospital of Padova, 35128 Padua, Italy; 6Institute of Radiology, University-Hospital of Padova, 35128 Padua, Italy; 7Department of Radiation Oncology, Veneto Institute of Oncology IOV-IRCCS, 35128 Padua, Italy; 8Pathology Unit, University-Hospital of Padova, 35128 Padua, Italy

**Keywords:** adrenocortical carcinoma, multidisciplinary team, overall survival, progression-free survival, personalized treatment

## Abstract

**Simple Summary:**

Adrenocortical carcinoma (ACC) is a rare and malignant disease with a poor prognosis in advanced disease. The complexity of ACC management requires a multidisciplinary approach, an emerging model of treatment, suitable for oncological diseases. Our study aims to determine the role of multidisciplinary team evaluation in affecting the overall survival and progression-free survival in ACC patients. We perform a retrospective analysis of ACC patients treated in Padova, with particular reference to their discussion in the multidisciplinary group of adrenal disease, definitively established in 2013 and defined as a collegial meeting between physicians involved in adrenal diseases. We describe a positive impact on survival rates in ACC patients after the multidisciplinary meeting.

**Abstract:**

We aimed to evaluate the role of adrenal multidisciplinary team evaluation (MTE) in affecting the overall survival (OS) and progression-free survival (PFS) in patients with adrenocortical carcinoma (ACC). We included in a retrospective monocentric study 47 patients with ACC. We divided our cohort into group 1 (without adrenal-MTE discussion, ACC diagnosis from 2004 to 2012, *n* = 14) and group 2 (diagnosis and beginning of treatments after 2013, all discussed in the adrenal MTE, *n* = 33). OS was defined by the survival between the first and the last visit, while PFS as the time from the first visit to the progression of the disease. Kaplan–Meier curves were used to compare OS and PFS between Group 1 and Group 2. Group 1^stages III–IV^ (*n* = 10) presented a shorter median OS than Group 2^stages III–IV^ (25 patients, 4 vs. 31 months, *p* = 0.023). Likewise, the median PFS was lower in Group 1 as compared to Group 2 (2.9 vs. 17.2 months, *p* < 0.001). The gain in PFS (6 months) was also confirmed in stage III-IV patients (2.9 vs. 8.7 months, respectively, for Group 1 and Group 2, *p* = 0.02). Group 1 presented a median PFS of 4 months, while the median PFS of Group 2 was 14.7 months (*p* = 0.128). In conclusion, we found a significant gain in terms of survival in patients after the MTE discussion in 2013. Therefore, ACC patients should be referred to a tertiary center, ideally from the time of diagnosis, to promptly apply all available treatments, according to the single patient’s clinical history and based on multidisciplinary management.

## 1. Introduction

Adrenocortical carcinoma (ACC) is a rare and malignant disease with an incidence of 0.7–2 per million population/year and a poor prognosis: the median overall survival (OS) is 3–4 years, reduced to 15 months in advanced or metastatic disease [1,2,3]. The peak of incidence is between 40 and 60 years and women are more often affected (55–60%). In adults, the majority of ACC are sporadic [1,2,3,4]. Germline mutations of TP53 characterize 50–80% of children with ACC and 4% of adult patients; on the other hand, somatic mutations of TP53 are observed in more than 50% of adult ACC patients, associated with a more aggressive phenotype [5,6]. Other somatic mutations described are the overexpression of IGF-2, the constitutive activation of the Wnt/Beta-catenin pathway, or the overexpression of steroidogenic factor 1 (SF-1) [7,8,9].

The clinical presentation of ACC is related to the hormonal excess in 50–60% of cases (hypercortisolism in up to 70% of cases, or excess of adrenal androgens in females), abdominal pain, nausea, vomiting, or cancer-associated symptoms (weight loss, fatigue, fever, night sweats) [1,2,3]. However, 10–15% of ACC are diagnosed within the group of incidental adrenal masses [10,11].

After initial endocrine workup and radiological staging of the disease [2,3], the management of patients with ACC should benefit from a multidisciplinary approach composed of a team of experts, as indicated in the European Society of Endocrinology guidelines, in collaboration with the European Network for the Study of Adrenal Tumors [2].

The treatment of ACC is challenging due to the rarity and malignancy of the disease. Complete surgical resection is the best option of cure in cases that are amenable to radical resection [12], and should be performed by surgeons experienced in adrenal and oncological surgery. The adjuvant treatment approved for ACC is mitotane, a parent compound of the insecticide dichlorodiphenyltrichloroethane (DDT), a lipophilic drug that exerts a strong effect on steroidogenesis by the inhibition of gene transcription of many steroidogenic enzymes with an adrenolytic effect. The intracellular target has not been identified yet; however, it is responsible for mitochondrial damage that leads to an apoptotic process. The therapeutic index of this drug is low; therefore, mitotane plasma levels must be periodically monitored (plasma concentration between 14 and 20 mg/L is considered the adequate balance between effectiveness and adverse events [1,2]).

In advanced metastatic ACC, first-line treatment involves combined chemotherapy with Cisplatin-Doxorubicin-Etoposide (EDP). Rescue therapy in case of disease progression after first-line treatment is possible, but with modest benefits in terms of activity and efficacy [1,2,3]. In recent years, some reports have highlighted the possibility of local treatment such as external beam radiotherapy, radiofrequency ablation, microwave ablation or chemoembolization in case of oligoprogression, but the available data are still limited [2,13,14].

Immunotherapy may offer a new approach in selected patients, because it has demonstrated a low-rate disease response. Recent data suggest that combining other treatments such as mitotane may enhance the efficacy of immunotherapy [15,16]. By far, there are no standardized indications for the application of these treatments in recurrent and/or advanced ACC, and therapeutic options for advanced ACC often result in a poor prognosis [2,4]. The complexity of the disease requires a multidisciplinary approach that is an emerging model of treatment for rare or oncological diseases [17]. The multidisciplinary team is used in several healthcare settings; however, it is difficult to demonstrate its efficacy due to methodological limitations and a lack of control groups, as reported by a systematic review of 12 studies regarding the relationship between a multidisciplinary approach to cancer and increased survival [18].

Recently, Daher et al. reported in a large study that a multidisciplinary approach to treatment is critical in optimizing outcomes of ACC patients because of the likelihood of a complete resection or the chance to receive more lines of systemic therapy [4]. However, to the best of our knowledge, the impact of an established multidisciplinary team evaluation (MTE) on clinical practice, with a direct comparison of survival before and after MTE approach in patients with ACC, has not been reported so far. Over the years, the University-Hospital of Padova and the Veneto Institute of Oncology have organized an adrenal disease multidisciplinary group (adrenal MTE) which has taken its final definition in 2013 (as reported in the graphical abstract).

The objective of our study was to evaluate the benefit in terms of survival (either OS or progression-free survival (PFS)), obtained through the multidisciplinary discussion of clinical cases of ACC, comparing the results to the survival of patients in the pre-Adrenal MTE era.

## 2. Materials and Methods

We performed a retrospective analysis on the ACC patients treated at our hospital, focusing on whether they benefited from the MTE. Data included all patients with a histological diagnosis of ACC since 2004; clinical, endocrine, oncological, and radiological data are available in the web-based database of the University-Hospital of Padova, used as an electronic Case Report/Record Form.

We used the European Network for the Study of Adrenal Tumors staging system: Stage I ACC < 5 cm in size and confined to the adrenal gland, without disease in nearby lymph nodes or distant sites (N0 and M0); Stage II ACC was defined as an N0M0 tumor > 5 cm confined to the adrenal gland; Stage III ACC was defined as a tumor with the disease in nearby nodes (N1), infiltration of surrounding tissue, or vascular extension without evidence of distant metastasis; Stage IV ACC was defined as a metastatic tumor (M1).

Adrenal MTE was initially fully established in 2013. It consists of a collegial discussion regarding adrenal diseases by a team of different experts, such as endocrinologists, oncologists, endocrine surgeons, radiologists, nuclear medicine physicians, pathologists, and radiation oncologists who take part in the discussion. All physicians, also outside of the MTE, can propose a case to the case-manager physician, who collects anamnestic/clinical/radiological data and prepares the case in a web-based platform (ad hoc prepared for all multidisciplinary meetings in our Healthcare provider). The MTEs are organized with periodic deadlines (every 3–4 weeks) and, at the end of each meeting, the final medical reports of the collegial discussion are digitally signed by all participants and uploaded to the local web-based database. We manage ACC patients according to the most recent guidelines [2,3]. The MTE discussions are tailored according to the clinical history of the patient, usually the first is performed immediately before surgery of a suspected patient, the second after the histological report (to plan mitotane and/or EDP, if needed, as well as further imaging), and then shortly after each treatment (second-/third-line chemotherapy, radiotherapy (RT), surgery for metastasis and so on) to evaluate the effectiveness of the treatment itself. Mitotane treatment was started in all patients, according to guidelines [2,3]. Mitotane concentrations were assessed monthly until the proposed range of efficacy, then every 2 to 4 months, according to the patient’s levels, and retrieved from the Lysosafe Online^®^ database, available at www.lysosafe.com (accessed on 7 February 2022), a free-of-charge service since 2005 with a gas chromatography/mass spectrometry method. EDP is the first-line treatment proposed for patients with advanced disease, according to guidelines. Moreover, after the MTE, we propose off-label treatments in compassionate use (according to the Italian Medicine Agency). Finally, we re-evaluate all imaging examinations not performed in Padova and measure steroids (especially cortisol) with mass-spectrometry.

We divided our cohort of ACC patients into two different groups: Group 1 (before adrenal MTE) included patients whose initial diagnosis and the greatest number of treatments (>75%) were performed from January 2004 to December 2012. Group 2 (post adrenal MTE) consisted of patients with initial diagnosis and beginning of treatments after January 2013. We selected to consider only newly diagnosed ACC patients after 2004, because the electronic records in Padova started in 2004; therefore, clinical history and medical reports were full available. Moreover, the time span of the two groups was similar (nine years).

We compared the two groups considering gender, age at diagnosis, incidental finding of the ACC, endocrine secretion (cortisol, androgens, or both), first-line and second-line treatments, RT, surgery of primitive tumor, re-intervention for metastasis, stage and the total number of second-line treatments. We considered as second-line treatments all treatments except surgery of the primary tumor, mitotane and EDP.

The primary endpoint was to determine the role of the MTE in affecting OS and PFS in ACC patients who arrived at our attention during their diagnostic course. OS was calculated from the date of diagnosis to that of the last follow-up visit available (until December 2021) or death. We also performed a further explorative analysis of OS considering time from the first visit at our institution, since some patients were referred with an already established diagnosis of ACC, performed in other centers. PFS was calculated as the time from the first treatment (including surgery) to the appearance of disease progression. We also performed a sub-analysis of PFS, named PFS^first visit^, considering the time from the first visit to our institution to the appearance of progression disease.

We performed a descriptive analysis using frequencies, means, and dispersion measures. Data between the two groups were compared using the X^2^ test for qualitative variables (or Fisher’s exact test when the cell count was <5) and the T-test for quantitative ones. A *p*-value <0.05 was considered statistically significant (two-sided tests). Alive patients were censored at the date of the last follow-up. Survival analysis was performed by Kaplan–Meier curves median and standard error of survival; OS and PFS were compared between the two groups using the log-rank test. The statistical analysis was performed using SPSS software version 24.

## 3. Results

### 3.1. Description of Patients with ACC

Since 2004, 67 patients with histological confirmation of ACC have been considered; 20 patients were excluded (17 without sufficient clinical data or lost in follow-up, 3 patients were newly diagnosed in 2022). According to inclusion criteria, 47 patients were analyzed. In the whole cohort, the mean age at diagnosis was 52 years (range 19–79), 25 subjects were females (53%), and endocrine activity (hypersecretion of cortisol, androgens, or both) was diagnosed in 27 patients (57%). In the entire population, the median follow-up was 68 months (Confidence Interval (CI) 95% 24.4–111.6), similar in stage III-IV patients (*n* = 35) (68 months, CI 95% 2.9–133.1).

Group 1 was made up of 14 patients (28% females); 10 were stage III-IV (71%), mean age at diagnosis was 56.3 years (range 26–79). Group 2 consisted of 33 patients (63% females); 25 of them presented with stage III-IV (75%), mean age at diagnosis was 50.2 years (range 19–78). Group 1 and Group 2 were similar in terms of age at presentation, incidental finding of ACC, hormonal secretion, stage distribution, chemotherapy with EDP, RT, cytoreductive surgery of the primary tumor, secondary surgery for metastasis, and the total number of second-line treatments, as reassumed in Table 1.

Among patients who underwent second-line treatments (7/14 patients in Group 1 and 14/33 patients in Group 2), the mean number of treatments in Group 1 was 1.71 (standard deviation of 1.11), while it resulted superior in Group 2 (2.5; standard deviation of 2.22), although not significantly different (*p* = 0.292).

### 3.2. Survival Analysis

Considering stage III-IV patients (*n* = 35), the mean OS was 28 and 88 months for Group 1 and Group 2, respectively (*p* = 0.006; median OS 4 months (95% CI 1.9–6)), and 145 months (95% CI 0–294) for Group 1 and Group 2, respectively, reported in Figure 1.

In 35 patients with stage III–IV disease, we also analyzed the OS at the time of registration of the first visit at our institution and the last visit available (because some patients were referred to Padova after an already-established diagnosis of ACC made in different units). In this setting, Group 1 presented a lower median OS (4 months; 95% CI 1.9–6) compared to Group 2 (31 months; 95% CI 1.2–60.7, *p* = 0.023; Figure 2).

Regarding PFS, we observed a median PFS^first treatment^ in Group 1 of 2.9 months (95% CI 0–6.1), compared to a median PFS^first treatment^ of 17.2 months (95% CI 3–31.4) in Group 2 (*p* < 0.001). The median PFS ^first treatment^ was 2.9 months (95% CI 1.5–4.2) in Group 1 stage III-IV and 8.7 months (95% CI 2.3–15, *p* = 0.02) in Group 2 stage III-IV (Figure 3).

We also performed a PFS analysis considering the time from the first visit to our center to the appearance of progression disease (PFS^first visit^). Group 1 presented a median PFS^first visit^ of 4 months (95% CI 0–8.4), while the median PFS^first visit^ was equal to 14.7 months (95% CI 7.9–21.5) in Group 2 (*p* = 0.128). In stage III-IV patients, the median PFS^first visit^ was 4.1 months (95% CI 0.3–7.7) and 9.6 months (95% CI 2.3–16.8), respectively, in Group 1 and Group 2 (*p* = 0.915, Figure 4).

## 4. Discussion

ACC is a rare malignancy with a poor prognosis and a high risk of recurrence. Systemic therapies have demonstrated low response rates and, currently, there are no standardized treatments for advanced and recurrent forms of ACC due to the lack of randomized and prospective trials [19]. The rarity and complexity of the disease highlight the need for a multidisciplinary evaluation, including experts in adrenal disease [11,20,21].

The multidisciplinary approach provides standardization of treatments, allowing personalized patient management with improved outcomes. A rare and complex disease like ACC should be treated in tertiary centers, in order to optimize the management with expertise [2,22]. Moreover, a shared approach can be proposed with teleconsultation, useful in the recent SARS-CoV-2 pandemic waves [23].

Regarding the effect of MTE on measurable outcomes, in terms of OS and PFS, we analyzed the survival rates in two groups of ACC patients, before and after the adrenal MTE. We found an improvement (in terms of prolonged survival) in patients who were taken in charge since 2013 (the year of the establishment of adrenal MTE), both considering the delay of diagnosis and the time of the first visit to the MTE (usually Endocrine Unit, Oncology Division or Endocrine Surgery). In particular, regarding patients with advanced disease (stage III-IV ACC) and considering the time from diagnosis, Group 2 (patients who were treated after an MTE) presented a median OS greater than those of Group 1 (patients treated without MTE discussion). It seems that patients in Group 1 presented an unbalanced outcome with Group 2. Nonetheless, some patients arrived at our attention late after the first diagnosis, because of a new recurrence or need for treatments available only in tertiary centers. Therefore, it is more correct to describe the time from the first visit to our center (immediately before the MTE) to the last available visit. Considering the time of registration for the first visit at our institution, and the stage III-IV, Group 2 had a median OS rate of 31 months as compared to 4 months for Group 1.

A recent multicentric study in the United States found improved OS in a cohort of 2886 ACC patients diagnosed from 2004 to 2016, focusing on centralization of care at tertiary centers, compared to peripheral ones, especially for patients diagnosed between 2010 and 2016 [24]. A 30% reduction in the hazard of death in a tertiary center was observed. The authors explained this finding by significant differences in surgical management, rate of positive surgical margins, and the use of systemic therapies. Moreover, they explained the significant difference in OS rates due to the multidisciplinary structure of tertiary centers as compared to the individual hospital’s case volume. The finding of the greater survival benefit after 2010 was thought to be secondary to the emerging research on the use of mitotane [25], especially in the adjuvant setting [26], and the corresponding increase in the use of adjuvant systemic therapies.

Another monocentric study on 330 ACC patients in a tertiary center reported a median OS in all stages of 38 months. The authors analyzed the median OS dividing patients by stage, showing a median OS of 42 months and 11 months, respectively, in stage III and IV patients [27]. These data are comparable to our findings regarding the OS of Group 2. Indeed, this latter subset of patients reflects the real potential of a tertiary center, since patients who belong to this group were treated by a referral center from the initial diagnosis of ACC.

A Dutch study conducted a retrospective and population-based survival analysis comparing the OS after surgery in 189 ACC patients treated in Dutch Adrenal Network (DAN) hospitals. The DAN was established with the aim to improve ACC patient care by a collaboration among specialists from different centers; it consists of a multidisciplinary meeting of experts who organized discussion of clinical cases and ideas for research programs. The authors found a longer OS in operated patients in DAN hospitals in comparison to those treated in non-DAN hospitals (*p* = 0.044). The median OS was 22 months and 6 months, respectively in stages III and IV [28].

The importance of achieving a radical resection after surgery (R0 margins) by performing adrenal surgery only in referral centers has been highlighted for a long time. Recently, a multicentric cohort study in 128 metastatic ACC patients has compared the OS rates between a group of patients who underwent primary tumor resection before systemic therapy and those without primary tumor resection. The results showed that cytoreductive surgery of the primary tumor in patients with metastatic ACC is associated with prolonged survival (HR 3.18; 95%CI: 2.34–4.32) [12].

Regarding the total number of treatments, we found no difference between the number of therapies performed over the years in the two groups, even since 2013 with the establishment of adrenal MTE. Despite this, the survival rate was increased in Group 2, probably thanks to the higher expertise and the better quality of treatments achieved over the years, with correct timing of application, in a context of personalized therapy.

Our analysis of PFS rates showed that, when considering the relapse time from first treatment to progression disease, there is a significant difference between Group 1 (median PFS 2.9 months) and Group 2 (median PFS 17.2 months), maintained also considering only patients at the III-IV stages. On the contrary, PFS rates in the two groups were similar considering the time since the first visit to our center to disease progression. In our opinion, this can be due to the delayed referral to a tertiary center regarding some cases with more severe or advanced diseases. Some patients are centralized late in their diagnostic-therapeutic course when disease progression is already present and the timing of correct treatment is no longer applicable.

## 5. Conclusions

Despite its strengths, our work presents some limitations. First, it is an observational study, with a reduced number of patients (reflecting the low incidence of ACC), and a controlled or a randomized study was not feasible. The expertise of multimodal treatment in patients with ACC started in Padova decades ago, since it is an academic referral center. However, in the last years, the availability of novel treatments (such as immunotherapy), the gain in novel facilities (imaging, laboratory assay), and the increased expertise of physicians (also through MTE) can explain prolonged survival. Moreover, we acknowledge that the best control group is the study of an independent cohort in another referral center that does not perform an MTE. It was not feasible in our case (Padova covers a large portion of Northern Italy), and a multicenter study can address this issue.

In our experience, MTE positively impacts survival rates in patients whose therapeutic course benefited from the application of decisions taken by multidisciplinary adrenal discussion. It seems implicit that in many academic settings the presence of multimodal treatment, supportive care, and facilities can improve patients’ survival, and that ACC patients need an MTE (as proposed by guidelines). According to our work, we can confirm that MTE is effective: ACC patients must be referred to a tertiary center, ideally from the time of diagnosis (rapid management provides better survival outcomes), because this is a rare and aggressive disease for which the available treatments have to be promptly applied, according to the single patient. Our work should encourage sanitarian stakeholders to create networks for patients.

## Figures and Tables

**Figure 1 cancers-14-03904-f001:**
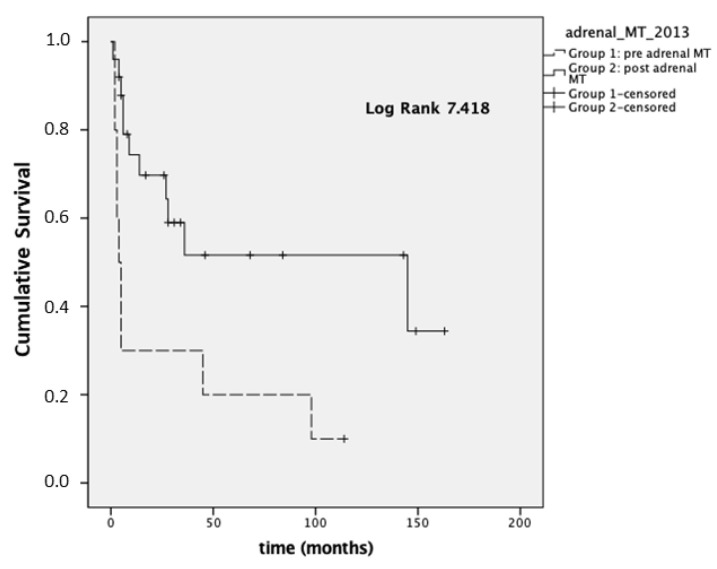
Overall survival (OS) in stages III–IV.

**Figure 2 cancers-14-03904-f002:**
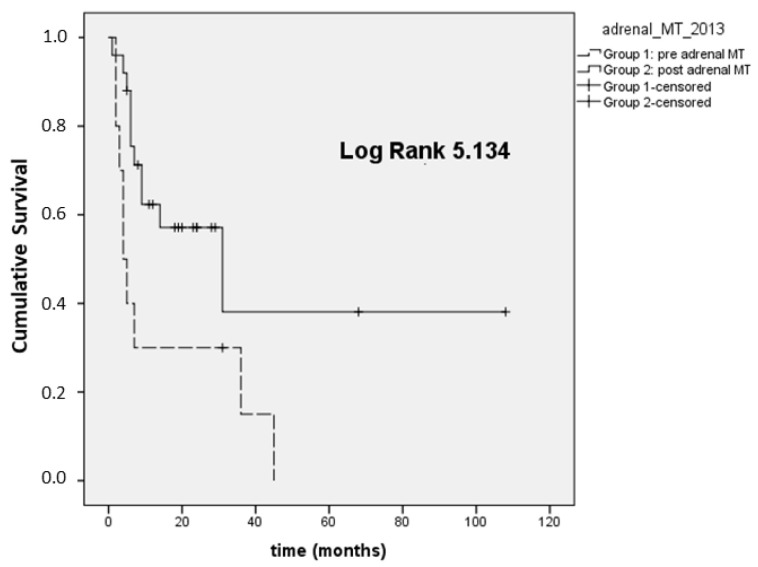
Overall survival ^first visit-last FU^ (OS ^first visit-last FU^) in ACC stage III–IV.

**Figure 3 cancers-14-03904-f003:**
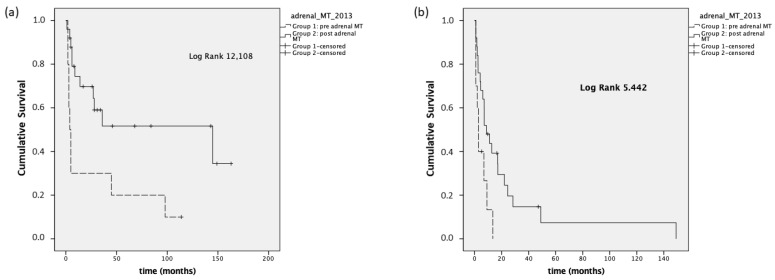
Panel (**a**): PFS ^first treatment^, stage I–IV; panel (**b**): PFS ^first treatment^, stage III–IV.

**Figure 4 cancers-14-03904-f004:**
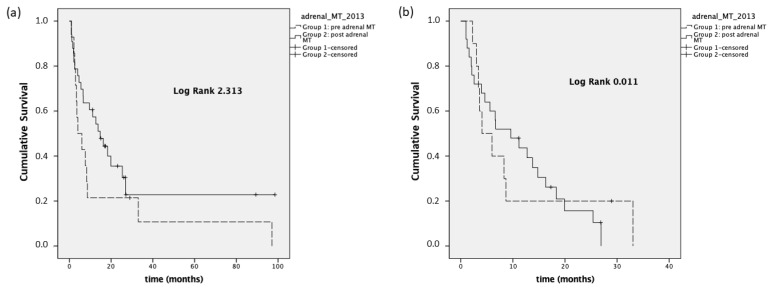
Panel (**a**): PFS^first visit^, stage I–IV; panel (**b**): PFS^first visit^, stage III–IV.

**Table 1 cancers-14-03904-t001:** Descriptive analysis of the two groups.

	Group 1 (Pre-Adrenal MTE)	Group 2 (Post-Adrenal MTE)	*p*
gender (% females)	28%	63%	0.053
age at diagnosis (mean)	56.29	50.24	0.230
incidental finding yes/no	2/14	11/20	0.178
secretion yes/no	9/14	18/33	0.093
EDP yes/no	7/14	17/33	1.000
second-line treatments yes/no	7/14	14/33	0.752
RT yes/no	2/14	7/33	0.704
primary surgery yes/no	10/14	29/33	0.215
secondary surgery for metastasis yes/no	4/14	8/33	0.731
stage I-II	4/14	8/33	0.745
stage III	1/14	5/33	
stage IV	9/14	20/33	
total *n*° of second-line treatments (mean)	1.71	2.5	0.292

## Data Availability

Data are available upon request to the corresponding author.

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
