# Peer review of "The Overall Survival and Progression-Free Survival in Patients with Advanced Adrenocortical Cancer Is Increased after the Multidisciplinary Team Evaluation"

_cancers, 2022, doi:10.3390/cancers14163904_

Round 1
Reviewer 1 Report
The article of Irene Tizianel et al. entitled “The overall survival and progression-free survival in patients with advanced adrenocortical cancer is increased after the multidisciplinary team evaluation” is a retrospective study carried out on 47 operated patients for ACC.
This study aimed at investigating the role of multidisciplinary team evaluation (MTE) in patients with this rare poor prognosis cancer.
All included patients were divided into 2 groups depending on when the diagnosis and treatment was made (before or after 2013):
-group 1 (before MTE): 14 patients from which 10 were stage III-IV;
-group 2 (after MTE): 33 patients from which 25 were stage III-IV.
The 2 groups were compared.
The follow up was 68 months.
The objective was to determine the role of multidisciplinary team on survival and progression-free survival (OS and PFS).
In their analysis the authors found better results since their multidisciplinary team was established.
Considering the III-IV stage patients, the authors found that the OS was 28 and 88 months respectively in the Group 1 and 2.
The Group 1 had lower median OS (4 months vs 31 months).
The authors found a median PFS first treatment of 2.9 months in the Group 1 and 17.2 months in the Group 2; the median PFS first visit was of 4 months in group 1 vs 14.7 months in the Group 2.
The authors showed that there was a significant difference about the relapse time from first treatment to progression disease in the 2 groups (2.9 months vs 17.2 months; p<0.001). This is a very important finding.
The authors found better results after 2013 since the care of these patients was entrusted to a multidisciplinary team specialized in endocrinology, endocrine surgery and nuclear medicine.
This study is very interesting as it highlights the importance of a dedicated team for the management of this rare cancer.
Rapid management of this tumor is known to provide better disease-free survival outcomes.
For these raisons I support the publication of this manuscript.
Minor comment:
I recommend to specify the acronym “MT” respectively in “Page 3, last line of the paragraph introduction and Page 4 line 13.
The authors should always use the same acronym MTE, sometimes they use MT.
Author Response
Reviewer #1:
The article of Irene Tizianel et al. entitled “The overall survival and progression-free survival in patients with advanced adrenocortical cancer is increased after the multidisciplinary team evaluation” is a retrospective study carried out on 47 operated patients for ACC.
This study aimed at investigating the role of multidisciplinary team evaluation (MTE) in patients with this rare poor prognosis cancer.
All included patients were divided into 2 groups depending on when the diagnosis and treatment was made (before or after 2013):
-group 1 (before MTE): 14 patients from which 10 were stage III-IV;
-group 2 (after MTE): 33 patients from which 25 were stage III-IV.
The 2 groups were compared.
The follow up was 68 months.
The objective was to determine the role of multidisciplinary team on survival and progression-free survival (OS and PFS).
In their analysis the authors found better results since their multidisciplinary team was established.
Considering the III-IV stage patients, the authors found that the OS was 28 and 88 months respectively in the Group 1 and 2.
The Group 1 had lower median OS (4 months vs 31 months).
The authors found a median PFS first treatment of 2.9 months in the Group 1 and 17.2 months in the Group 2; the median PFS first visit was of 4 months in group 1 vs 14.7 months in the Group 2.
The authors showed that there was a significant difference about the relapse time from first treatment to progression disease in the 2 groups (2.9 months vs 17.2 months; p<0.001). This is a very important finding.
The authors found better results after 2013 since the care of these patients was entrusted to a multidisciplinary team specialized in endocrinology, endocrine surgery and nuclear medicine.
This study is very interesting as it highlights the importance of a dedicated team for the management of this rare cancer.
Rapid management of this tumor is known to provide better disease-free survival outcomes.
For these raisons I support the publication of this manuscript.
[Reply to reviewer #1]
We thank you for the positive review which has instilled confidence in us and helped us revise the article to make it more relevant for the readers of this esteemed journal.
Minor comment:
I recommend to specify the acronym “MT” respectively in “Page 3, last line of the paragraph introduction and Page 4 line 13.
The authors should always use the same acronym MTE, sometimes they use MT.
[Reply to reviewer #1] We corrected the manuscript and used the acronym MTE in the whole manuscript, also in abstract and tables.

Reviewer 2 Report
In this manuscript, the authors discuss the value of implementing a multidisciplinary program for the evaluation of patients with adrenocortical carcinoma. There are several flaws in the design of the research that need to be addressed:
1. The number of patients is low, particularly group 1, and statistical analysis and results may not be representative of the differences. Fisher exact test should be used rather than standard Xi square
2. There are several variables that can affect the outcome of patients in the two groups that should be addressed, e.g. experience of of oncology and nursing staff in primary centers versus tertiary centers, locations of treatment centers, urban versus rural, availability of immunotherapy, targeted therapy or other modern treatment in both centers, etc.
3. More information is needed about the length of follow-up and span of diagnosis and treatment, particularly in group 1. Patients diagnosed prior to 2013 have longer follow-up with clear outcome than those diagnosed after 2013. Differences between the two groups may be related to 2013 cut-off time point, as earlier times may carry different social standards and more primitive diagnostic and treatment modalities. A better comparison of survival analysis would be between primary care centers without MT program versus tertiary centers implementing the MT program, all carried out at the present time or similar time interval.
4. Details of the MT programs are needed. Simply discussing cases does not lead to patient improvement. Authors may want to explain details and algorithm of the MT programs, including steps of treatment, follow-up and novel diagnostic methods.
Author Response
Reviewer #2: In this manuscript, the authors discuss the value of implementing a multidisciplinary program for the evaluation of patients with adrenocortical carcinoma. There are several flaws in the design of the research that need to be addressed:
[Reply to reviewer #2] We sincerely appreciate the efforts put forth by the editors and reviewers in the review of our manuscript.
- The number of patients is low, particularly group 1, and statistical analysis and results may not be representative of the differences. Fisher exact test should be used rather than standard Xi square
[Reply to reviewer #2] Thank you so much for the suggestion. We have already reported the result of the Fisher test in the results; nonetheless, it was not clearly specified in the materials and methods section. In the new version of the paper, we have added it.
- There are several variables that can affect the outcome of patients in the two groups that should be addressed, e.g. experience of of oncology and nursing staff in primary centers versus tertiary centers, locations of treatment centers, urban versus rural, availability of immunotherapy, targeted therapy or other modern treatment in both centers, etc.
[Reply to reviewer #2] we can partially change our manuscript because Padova is a referral center, in an academic hospital, with long-lasting experience in ACC patients (first patients managed in 1980 and first ever publication by endocrinology of Padova in 1987). However, we agree with your suggestion and according also to other reviewers we add them in a limitation paragraph close to the conclusions of the manuscript.
- More information is needed about the length of follow-up and span of diagnosis and treatment, particularly in group 1. Patients diagnosed prior to 2013 have longer follow-up with clear outcome than those diagnosed after 2013. Differences between the two groups may be related to 2013 cut-off time point, as earlier times may carry different social standards and more primitive diagnostic and treatment modalities. A better comparison of survival analysis would be between primary care centers without MT program versus tertiary centers implementing the MT program, all carried out at the present time or similar time interval.
[Reply to reviewer #2] Thank for the insightful comment. However, a direct comparison with other centers that do not use multidisciplinary evaluation is not feasible, and we add it as a food-for-thought idea (a multicentric study with and without multidisciplinary evaluation) to address this issue.
- Details of the MT programs are needed. Simply discussing cases does not lead to patient improvement. Authors may want to explain details and algorithm of the MT programs, including steps of treatment, follow-up and novel diagnostic methods.
[Reply to reviewer #2] A partial description of the MTE evaluation was depicted in the introduction of the previous version. We add the required data in a novel part of the material and methods section, with a further description of the MTE.

Reviewer 3 Report
I believe that the paper was written in a clear and understandable way. The data are clearly presented and the conclusions appear to be shared.
I consider the paper to be of absolute value as it establishes management criteria for adrenal carcinoma capable of improving the outcome of this disease. Furthermore, the center where the research was carried out is of absolute value in the field of adrenal pathologies and therefore can fully speak of the pathology.
Author Response
Reviewer #3: I believe that the paper was written in a clear and understandable way. The data are clearly presented and the conclusions appear to be shared.
I consider the paper to be of absolute value as it establishes management criteria for adrenal carcinoma capable of improving the outcome of this disease. Furthermore, the center where the research was carried out is of absolute value in the field of adrenal pathologies and therefore can fully speak of the pathology.
[Reply to reviewer #3] We thank you for the positive review which has instilled confidence in us.

Reviewer 4 Report
Adrenocortical carcinoma (ACC) is a very aggressive malignancy, and the authors presented a very simple study design to differentiate ACC outcome based on overall survival (OS) and progression free survival (PFS), using only 47 available cases in their institution (University-Hospital of Padova and the Veneto Institute of Oncology), by dividing into “non-multidisciplinary team evaluation” before 2013 (N=14, including 10 stage III-IV) and “multidisciplinary team evaluation”after 2013 (N=33, including 25 stage III-IV). No significant differences in treatment protocols were presented related to the use of traditional therapy (surgery, chemotherapy, use and monitoring of the adrenolytic mitotane, and radiotherapy)or more advanced or novel categories.
It is already standard to have collegial discussion by a team of different experts, specially at highvolume centers of several American Centers and in European countries (particularly following the European Network for the Study of Adrenal Tumors - ENSAT guidelines) since many years. Such teams centralized complex decisions for personalized patient management. Better decisions about stage III and IV treatment options and supportive care, based on type of lesions, its site, and severity, usually come from an academic team (oncologist, endocrinologist, surgeon, and the pathologist) rather than a single professional. There is no doubt that multidisciplinary team (or academic) and facility type are associated with improved therapeutic outcomes and survival in ACC patients. This is implicit in many academic settings, no need to prove the obvious (as the authors wrote in the introduction: “However, to the best of our knowledge, the impact of an established multidisciplinary team evaluation (MTE) on clinical practice, with a direct comparison before and after MTE approach in patients with ACC has not been reported so far.”). Take for example the significant differences in practice patterns, based on the decision to more aggressive surgical treatment associated or not with a prolonged mitotane treatment by a team of surgeons and clinicians, or by a single professional in a hospital with less experience.
There are studies using a larger number of ACC cases showing differences in practice patterns and overall survival across types of treating Centers, all of them with better median OS as shown in the present study after 2013. Given the small number of cases, it is impossible to adjust for patient (gender, age at diagnosis), steroid levels, staging and other characteristics, or determining if the outcomes were more strongly correlated with specific decision differences. In summary, this is a study without novel data, probably suitable for a letter or short communication.
Author Response
Reviewer #4:
Adrenocortical carcinoma (ACC) is a very aggressive malignancy, and the authors presented a very simple study design to differentiate ACC outcome based on overall survival (OS) and progression free survival (PFS), using only 47 available cases in their institution (University-Hospital of Padova and the Veneto Institute of Oncology), by dividing into “non-multidisciplinary team evaluation” before 2013 (N=14, including 10 stage III-IV) and “multidisciplinary team evaluation”after 2013 (N=33, including 25 stage III-IV). No significant differences in treatment protocols were presented related to the use of traditional therapy (surgery, chemotherapy, use and monitoring of the adrenolytic mitotane, and radiotherapy)or more advanced or novel categories.
It is already standard to have collegial discussion by a team of different experts, specially at highvolume centers of several American Centers and in European countries (particularly following the European Network for the Study of Adrenal Tumors - ENSAT guidelines) since many years. Such teams centralized complex decisions for personalized patient management. Better decisions about stage III and IV treatment options and supportive care, based on type of lesions, its site, and severity, usually come from an academic team (oncologist, endocrinologist, surgeon, and the pathologist) rather than a single professional. There is no doubt that multidisciplinary team (or academic) and facility type are associated with improved therapeutic outcomes and survival in ACC patients. This is implicit in many academic settings, no need to prove the obvious (as the authors wrote in the introduction: “However, to the best of our knowledge, the impact of an established multidisciplinary team evaluation (MTE) on clinical practice, with a direct comparison before and after MTE approach in patients with ACC has not been reported so far.”). Take for example the significant differences in practice patterns, based on the decision to more aggressive surgical treatment associated or not with a prolonged mitotane treatment by a team of surgeons and clinicians, or by a single professional in a hospital with less experience.
There are studies using a larger number of ACC cases showing differences in practice patterns and overall survival across types of treating Centers, all of them with better median OS as shown in the present study after 2013. Given the small number of cases, it is impossible to adjust for patient (gender, age at diagnosis), steroid levels, staging and other characteristics, or determining if the outcomes were more strongly correlated with specific decision differences. In summary, this is a study without novel data, probably suitable for a letter or short communication.
[Reply to reviewer #4] Thank so much for the revision, we upload a revised version of the manuscript as indicated by the Editors. We agree that the better survival in an academic center is due to a high-load of cases (secondary to centralized complex decisions) and the presence of several facilities (novel oncological treatment, conventional and nuclear imaging) and high-expertise physicians. Our number of cases is reduced respect to other series, however it is a mono-centric experience. We add a novel paragraph, according also to other reviewers, with the main limitations of the study.

Round 2
Reviewer 4 Report
I believe that these authors have the potential to contribute with important studies on ACC in the future. However, the revised version and the authors' response are still not sufficient to change my previous evaluation. This type of study together with the small number of cases have limitations in novelty and in specific technical details capable to account for differences in outcome (other than PS and PFS).